# Copper-Catalyzed Homocoupling of Boronic Acids: A Focus on B-to-Cu and Cu-to-Cu Transmetalations

**DOI:** 10.3390/molecules27217517

**Published:** 2022-11-03

**Authors:** Aude Salamé, Jordan Rio, Ilaria Ciofini, Lionel Perrin, Laurence Grimaud, Pierre-Adrien Payard

**Affiliations:** 1Laboratoire des Biomolécules (LBM), Département de Chimie, Ecole Normale Supérieure, PSL University, Sorbonne Université, CNRS, 75005 Paris, France; 2Université de Lyon, Université Claude Bernard Lyon I, CNRS, INSA, CPE, UMR 5246, ICBMS, 1 rue Victor Grignard, 69622 Villeurbanne, France; 3École Nationale Supérieure de Chimie de Paris, Centre National de la Recherche Scientifique, Institute of Chemistry for Life and HealthSciences, PSL Research University, 75006 Paris, France

**Keywords:** transmetalation, Cu-catalysis, cross-coupling, homocoupling, mechanism, DFT, cyclic voltammetry, boronic acid

## Abstract

Controlling and understanding the Cu-catalyzed homocoupling reaction is crucial to prompt the development of efficient Cu-catalyzed cross-coupling reactions. The presence of a coordinating base (hydroxide and methoxide) enables the B-to-Cu(II) transmetalation from aryl boronic acid to Cu^II^Cl_2_ in methanol, through the formation of mixed Cu-(μ-OH)-B intermediates. A second B-to-Cu transmetalation to form bis-aryl Cu(II) complexes is disfavored. Instead, organocopper(II) dimers undergo a coupled transmetalation-electron transfer (TET) allowing the formation of bis-organocopper(III) complexes readily promoting reductive elimination. Based on this mechanism some guidelines are suggested to control the undesired formation of homocoupling product in Cu-catalyzed cross-coupling reactions.

## 1. Introduction

Metal-catalyzed coupling reactions are a staple of organic synthesis as they allow the controlled extension of the carbon backbone of molecules. One of the first historically described coupling reactions was reported by Ullman at the beginning of the last century (1901) to couple two identical aryl halides (homocoupling) in the presence of a stoichiometric amount of Cu(0) [1,2,3,4]. While Pd-catalyzed cross-coupling reactions are among the most used nowadays, Cu catalysis proved to be a nice complement [5] and is particularly interesting to catalyze C-C and C-heteroatom bond formations [5,6,7,8,9,10,11,12,13,14,15,16,17,18,19,20].

In the last decade, several protocols have been disclosed describing that mild conditions and low catalyst-loadings can perform the homocoupling of boronic acids using Cu salts (Figure 1A) [6,7,8,9]. Cu-catalyzed homocoupling of boronic acids takes place at room temperature in the presence of a weak base such as K_2_CO_3_ (Figure 1A) using Cu^II^Cl_2_ or Cu^I^Cl as catalyst precursors [7].

Controlling and understanding the Cu-catalyzed homocoupling reaction is of high interest (i) to enable efficient formation of symmetrical biphenyl compounds under mild conditions but also (ii) to disfavor this process in the case of Cu-catalyzed cross-coupling reactions. Deciphering the mechanism of homocoupling reactions is thus crucial to prompt the development of efficient Cu-catalyzed Cham-Lan [22,23,24,25,26] or Cu-catalyzed Suzuki-liked cross-couplings (Figure 1A) [10,11]. By analogy with the established mechanism for Pd-catalyzed oxidative coupling of boronic acids [27,28], it was proposed that the Cu-catalyzed homocoupling of boronic acid takes place through two successive transmetalations on Cu(II) followed by (i) direct reductive elimination to yield Cu(0) or (ii) one-electron oxidation into Cu(III) and subsequent reductive elimination to yield Cu(I) (Figure 1B) [6,7,8,9,21]. However, there is no evidence to support these mechanisms. Moreover, the role played by the base was not fully clarified. Indeed, starting from Cu(I), no base is required for the reaction to proceed [15], but the addition of a base is essential starting from Cu(II) [7]. Herein, we report evidence for an alternative homocoupling reaction pathway involving successive B-to-Cu and Cu-to-Cu transmetalation steps.

The mechanism of the Cu^II^Cl_2_-catalyzed homocoupling reaction of arylboronic acids in methanol under inert atmosphere was investigated experimentally using a combination of NMR, cyclic voltammetry, conductimetry pH-metry, and kinetics monitoring. The experimental insights were complemented by the exploration of reaction pathways computed at the DFT level. In particular, the effect of the added base and the involvement of higher-order Cu intermediates are clarified.

## 2. Results and Discussion

### 2.1. Cu-Catalyzed Homocoupling Reaction: Effect of Cu on Selectivity

The stoichiometric reaction of Cu^II^Cl_2_ (10 mM) with 2 equiv of *para*-fluorophenylboronic acid (*p*-F-C_6_H_4_)B(OH)_2_ (**I**) was monitored using ^19^F NMR in MeOH (Figure 2). The NMR tube was prepared under inert atmosphere to avoid the oxidative regeneration of Cu(II) from reduced Cu complexes and, thereof, the subsequent oxidative coupling (Appendix A). In MeOH, in the absence of an added base, (*p*-F-C_6_H_4_)B(OH)_2_ (−112.2 ppm) is stable and no decomposition reaction was observed after 5 h at r.t. (Appendix A).

Tetrabutylammonium hydroxide (TBAOH) was used as a model base for this reaction since it is soluble in methanol and allows us to separate the effect of the base from the ones related to a coordinating counter-cation. In the presence of 20 equiv of (*p*-F-C_6_H_4_)B(OH)_2_ and 12 equiv of TBAOH, the signal of the boronic acid reagent (**I**) decreased rapidly and several new signals appeared at −117.7, −115.1, −126.5, and −127.9 ppm. They correspond to the homocoupling product **II**, the product of the boronic acid protodemetalation **III**, the phenol **IV**, and the ether **V** oxidation products of the boronic acid, respectively (Figure 2 and Appendix A). All signals were attributed by addition of a small amount of each pure product in the NMR tube.

Since the boron derivative is stable in the absence of Cu^II^Cl_2_, even in the presence of a base (Appendix A), protodemetalation is expected to be mediated by Cu. Consequently, the formation of a protodemetalation product is indicative of B-to-Cu transmetalation followed by protonation, while the formation of homocoupling product comes from transmetalation and subsequent reductive elimination.

The ratio of protodemetalation vs. homocoupling products was shown to be highly dependent on the Cu^II^Cl_2_ and added base concentrations. At high Cu^II^Cl_2_ concentration (20 mM), no protodemetalation product was detected, while at low concentration about 20% of the protodemetalation product was observed (Appendix A). This indicates that the kinetics of the homocoupling reaction follows a higher reaction order with respect to Cu(II) than the kinetics of the protodemetalation reaction. This suggests the possible involvement of multinuclear Cu(II) complexes. The concentration of TBAOH was shown to impact the final product distribution: a high base concentration (0.8 equiv with respect to PhB(OH)_2_) favors the proto-demetallation (Appendix A).

### 2.2. Cyclic Voltammetry Monitoring of the Homocoupling Reaction

Electrochemical analysis methods were used to monitor the structure of ligandless Cu species under reaction conditions. Cyclic voltammetry is a powerful tool to access the oxidation state of a metal in solution [29] and, consequently, to scrutinize the oxidation state of a catalyst during oxidative or reductive chemical steps [27]. Cu^II^Cl_2_ (2 mM) is characterized in methanol (Figure 1A) by two reduction peaks at 0.25 and −0.75 V vs. the standard calomel electrode (SCE). They respectively correspond to the reduction to Cu(I) and subsequent deposition of Cu(0) at the electrode. The deposited Cu(0) is re-oxidized to Cu(I) at 0 and 0.2 V vs. SCE and Cu(I) to Cu(II) at 0.5 V vs. SCE. The addition of phenylboronic acid does not modify the shape of the voltammogram (Appendix A).

Upon addition of TBAOH (6 equiv) in the presence of PhB(OH)_2_ (10 equiv), the reduction peak related to Cu(II) disappears progressively (Appendix A), while the oxidation peak Ox3, proportional to the concentration of Cu(I), is increasing (Figure 1B). This highlights the formation of Cu(I) and the consumption of Cu(II). In addition, since the reduction in Cu(I) can only originate from reductive elimination, the variation of the concentration of Cu(I) with time relates to the kinetics of the coupling reaction.

To gain more insights into the role played by Cu and the base in the coupling mechanism, we monitored the kinetics of Cu(I) formation using a rotating disk electrode (RDE) [30]. Using this set-up, the potential of the working electrode can be imposed at 0.7 V vs. SCE to enable Cu(I) oxidation. In this condition, the current intensity measured is proportionated to the concentration of Cu(I) in the solution. In the presence of an excess of phenylboronic acid and of base, all kinetic profiles could be fitted by a pseudo first-order law in Cu(I) (Appendix A).

The variation of the apparent rate constant for the formation of Cu(I) with increasing pH is depicted in Figure 2. At low base concentration, the rate of the coupling reaction increased linearly with the concentration of added base, while at higher concentration (6 to 8 equiv), the rate of the reaction decreased sharply. By analogy with the proposed transmetalation mechanism, this bell-shape behavior could be indicative of the formation of a μ-OH-bridged Cu-(μ-OH)-B pre-transmetalation intermediate [31]. In the absence of base, the reaction is apparently limited because of the absence of OH groups, while at high concentration the predominance of Cu hydroxide and boronate species likely hamper the formation of mixed complexes between Cu and B.

### 2.3. Electronic Effects on the Kinetics of the Reaction: Hammett Plot

To get more information related to the electronic effects over the rate-determining state (RDS) of the overall coupling process, the kinetics of the Cu(I) formation was monitored for a series of *para*-substituted phenylboronic acid derivatives. The variations of the pseudo first-order rate constant were obtained at pH = 11 and 12. In both cases, electron-rich boronic acid derivatives react faster than electron-poor ones (Figure 3 and Appendix A). The negative slope of the Hammett plot for the reaction is thus indicative of a decrease in the partial charge on the aromatic ring during the RDS. This is consistent with transmetalation being the rate-determining step. The nonlinearity of the Hammett plot drawn can be indicative of a shift in pre-equilibrium or a change of the RDS step [32].

### 2.4. Structure of Cu(II) in Methanol

To interpret the effect of the base concentration on the kinetics of the coupling reaction, we attempted to gather more information regarding the structure of Cu^II^Cl_2_ in MeOH solution as a function of the pH. The conductivity of the Cu^II^Cl_2_ electrolyte in MeOH was studied as a function of the square root of the concentration (Appendix A). The nonlinear plot obtained is typical of a weak electrolyte resulting from a limited dissociation of Cu^II^Cl_2_ into CuCl^+^ and Cl^−^ in MeOH, in contrast to the behavior reported in water [33]. This agrees with the much lower permittivity of MeOH compared to water [34].

As PhB(OH)_2_ is a weak acid-in-methanol solution accordingly to Equation (1) (pKa_(MeOH)_ (PhB(OH)_2_/PhB(OH)_3_^−^) = 11.2, see Appendix A), the pH of the methanol solution can be controlled by adding sub-stoichiometric amounts of TBAOH to PhB(OH)_2_. Thereof, the resulting mixture of boronate PhB(OH)_3_^−^ and boronic acid acts as a pH buffer.
PhB(OR)_2 (MeOH)_ + ROH _(MeOH)_ = PhB(OR)_3_^−^
_(MeOH)_+ H^+^
_(MeOH)_; R = H, Me(1)

The [PhB(OH)_3_^−^]/[PhB(OH)_2_] concentration ratios and the associated pH range that enable the coupling reaction to proceed (Figure 2 and Appendix A) were estimated between pH 10.5 and 12.5. Under these conditions, the number of anionic ligands on Cu was determined using the Pourbaix diagram of Cu^II^Cl_2_ in MeOH. This diagram was constructed by measuring the open-circuit potential (OCP) of (i) an equimolar solution of Cu^I^Cl and Cu^II^Cl_2_ (2 mM/2 mM, top plot) at a Pt working electrode and (ii) a Cu(0) working electrode immersed in a 2 mM solution of Cu(I), as a function of the pH of the solution, which was varied by the addition of a small quantity of a 1 M solution of TBAOH in MeOH (Figure 4).

For pH ranging from 4 to 8, the potentials of the Cu(II)/Cu(I) and Cu(I)/Cu(0) couple show very little variation; this indicates that counter anions are not coordinated to either Cu(I) or Cu(II). Above pH 8, the Cu(II)/Cu(I) curve shows a nonlinear transition that is indicative of the partial formation of either [Cu^II^(OR)Cl] and/or [Cu^II^(OR)]^+^ complexes (pKa_1_ (CuCl_2_/Cu(OH)Cl) = 9). This transition is also indicated by the formation of green complex. Above pH 10, the slope of the later increases to a value of 113 mV/pH; this corresponds to the coordination of anionic ligands to Cu(II) to form the [Cu^II^(OR)_2_] (R = H, Me) alkoxide complexes that partially precipitate (pKa_2_ (CuCl_2_/Cu(OR)_2_ = 16.3). Of note, such a precipitate was never observed under the reaction conditions, even at a higher base concentration. For pH above 11, [Cu^I^(OH)] becomes the predominant species for Cu(I).

### 2.5. Computational Mechanistic Investigation: B-to-Cu(II) Transmetalation

Based on the results described in Section 2.4, Cu(II) is likely to exist under the form of a [Cu^II^(OR)_2_] complex, with OR^−^ = HO^−^, MeO^−^, PhB(OH)_3_^−^ depending on the reaction conditions, for pH ranging between 9.5 and 12.5. However, at low pH, the formation of [Cu^II^Cl(OR)] or [Cu^II^(OR)]^+^ cannot be excluded. Thereof, the ability of all these species to promote B-to-Cu(II) transmetalation was evaluated by means of a computational mechanistic investigation performed at the DFT level using the B3LYP hybrid functional [35,36,37,38]. Hereafter, energies refer to Gibbs energies (∆*G*, kcal mol^−1^) estimated at 298 K and 1 atm upon full geometry optimization. Solvation by methanol has been modeled by means of a micro-solvation approach [39] that combines an explicit description of the first solvation sphere and an implicit representation of the bulk solvation effects by a polarization continuum model [40]. The fully detailed computational approach is available in Section 4, Materials and Methods. In the following sections, DFT-optimized structures are tagged by Arabic numerals. If the complex is cationic, a + is added (e.g., **5^+^**). For neutral complexes, the counter anion is specified by a superscript (e.g., **4^OH^**). The presence and the number of explicit solvent molecules are indicated by a superscript n = (e.g., **4^+,n=2^**), ranging from 0 to 2. Finally, when spin intercrossing occurs, i.e., in Section 2.6, the spin state (singlet *s*, doublet *d*, triplet *t*) of the complex is specified in superscript brackets (e.g., **9^MeO(t),n=0^**). Transition states are indicated using the prefix **TS-**.

The transmetalation mechanism starting from variously solvated heterobimetallic adducts [Cu^II^(PhB(OH)_3_)(MeOH)_n_]^+^ (**3^+,n^**) between phenylboronate (**1**) and [Cu^II^MeOH_2_]^2+^ (**2**) was investigated. Figure 5 depicts the most relevant Gibbs energy profile in the case of the di-solvated [Cu^II^(PhB(OH)_3_)(MeOH)_2_]^+^ (**3^+,n = 2^**) heterobimetallic adduct. The influence of solvation on the stabilities of **3^+,n^** (n = 0, 1 or 2) and on the B-to-Cu transmetalation energy barriers is discussed in the SI (Appendix A) [39].

Complex **3^+,n = 2^** features two bridging μ-OH groups between Cu(II) and B atoms and the four-membered ring atoms are nearly co-planar (Φ_Cu-O-B-O_ = 1.5°, Figure 6A). The formation of such a mixed Cu-(μ-OH)-B dimer as a key pre-transmetalation species mirrors previous reports on Pd [41,42,43] or Ni complexes [44]. Starting from complex **3^+,n=2^**, one of the bridging hydroxy groups is exchanged with the phenyl group via a torsion along the non-exchanging B-µ-O bond. This isomerism is endergonic by 11.6 kcal mol^−1^ and gives the pre-transmetalation intermediate **4^+,n=2^**. In **4^+,n = 2^**, the interaction between Cu and the Ph moiety has a pronounced σ character, as indicated by the open B-C*^ipso^*-C*^para^* angle of 171° and the short Cu-C*^ipso^* distance (2.22 Å) (Figure 6A). This pre-activation allows the easy transfer of the Ph group via the early transition state **TS-4^+,n = 2^**. Relative to **3^+,n = 2^**, transmetalation via **TS-4^+,n = 2^** requires us to overcome an overall Gibbs energy barrier of 16.2 kcal mol^−1^ to yield the transmetalated complex *trans*-[Cu^II^(Ph)(MeOH)_2_(μ-OH)B(OH)_2_]^+^ (**5^+^**). This is in good agreement with previous calculations related to boronate transmetalation reported on a parent Cu system [45]. From **3^+,n = 2^** to **TS-4^+,n=2^**, the main contribution to the energy barrier is the formation of pre-complex **4^+,n = 2^**. From cationic complex **5^+^**, coordination of a second phenylboronate molecule [PhB(OH)_3_]^−^ (**1**) along with the release of boronic acid leads to intermediate **6**. The large exergonicity of the reaction and the overall Gibbs energy barrier of ca. 16 kcal mol^−1^ are in line with experimental data.

Alternative pathways of transmetalation have been explored starting from neutral monosolvated Cu(II) complexes [Cu^II^(MeOH)(X)(μ-OH)_2_B(Ph)(OH)], X = Cl (**3^Cl,n = 1^**), MeO (**3^MeO,n = 1^**). The Gibbs energies of the computed profiles are gathered in Table 1. The presence of a chloride bounded to Cu merely affects the overall transmetalation barrier (17.2 kcal mol^−1^) compared to the cationic system (16.2 kcal mol^−1^). By comparison, methoxide is slowing down transmetalation as indicated by an overall energy barrier of 23.7 kcal mol^−1^. This could be one possible explanation for the observed pH dependence of the homocoupling kinetics.

The lower ability to transmetalate to hydroxo- or methoxy-copper complexes is also reflected in the structure of the pre-transmetalation intermediate **4^X,n = 1^**. Both **4^OH,n = 1^** and **4^MeO,n = 1^** show very little σ-character of the Cu-C bond with bond length of 2.4 Å and flat B-C^ipso^-C par angle of 175°. This is related to a lower donation from Cu to Ph in agreement with the higher partial charge at the Cu in **TS-4^MeO,n = 1^** (1.326 |e^−^|) compared to **TS-4^Cl,n = 1^** (1.256 |e^−^|) (Appendix A). Finally, the formation of transmetalation products **5^X^** is either endergonic (X = Cl) or isergonic (X = OH and MeO).

Based on these considerations, we have assessed the influence of solvent coordination on the energy barrier of the transmetalation step (Appendix A). Mono-solvated **3^MeO,n = 1^** is merely more stable than the solvent-free **3^MeO,n=0^** (∆G = −0.7 kcal mol^−1^); both are likely in equilibria in MeOH solution. The formation of the pre-complex **4^MeO,n = 0^** is facilitated compared to the mono-solvated case (+9.5 kcal mol^−1^) and the energy of the subsequent transmetalation transition state **TS-4^MeO,n = 0^** is 4.7 kcal mol^−1^ lower in energy than the one computed for the mono-solvated transition state **TS-4^MeO,n = 1^** (∆G^‡^ = 18.0 vs. 22.7 kcal mol^−1^, Appendix A). These specific solvent effects on transmetalation make the use of an explicit solvent model mandatory for the description of such systems. Indeed, the dynamics of solvation plays an important role in slowing down or facilitating the transmetalation, as suggested for other related systems [46,47,48].

The mechanism presented above accounts for the experimental data regarding the transmetalation but does not rationalize the formation of homocoupling products. One possible mechanism to explain the homocoupling involves a second B-to-Cu transmetalation of the phenyl group followed by a reductive elimination (Figure 1). The energy profile of this reaction pathway were computed starting from [Cu^II^(MeOH)(PhB(OH)_3_)Ph] (**6**) (see Appendix A). Activation energies related to a second transmetalation have been computed to at least 25.1 kcal mol^−1^. Based on the experimental conditions and the kinetic measurements, this reaction sequence cannot thereof account for the formation of homocoupling products, though it is commonly suggested.

### 2.6. Computational Mechanistic Investigation: Cu(II)-Cu(II) Transmetalation and Reductive Elimination

To rationalize the formation of homocoupling products, we have considered alternative reaction pathways in which the second transmetalation takes place between two copper centers. Such a mechanism would support the higher kinetic order with respect to Cu experimentally inferred for the homocoupling reaction. In the following section, we will only discuss the reaction sequence starting from the monomeric PhCu(S)OMe (**7^MeO^**) that results from the first transmetalation (Figure 7).

The second transmetalation starts with the dimerization of **7^MeO(d)^** in its doublet state (S = 1/2) to form the high-spin di-solvated Cu^II^-Cu^II^ dimer **8^MeO(t),n = 2^** (S = 1), whose formation is computed exergonic by 6.3 kcal mol^−1^. The release of two solvent molecules from **8^MeO(t),n = 2^** leads to an even more stable dimer triplet complex **8^MeO(t),n = 0^** (∆*G* = −16.1 kcal mol^−1^ relative to **7^MeO(d)^**). Torsion along the nonreactive Cu^II^-(μ-OH) bond enables the exchange between one bridging μ-MeO group and the Ph ring to yield **9^MeO(t),n = 0^** in its triplet state. This step is endergonic by 8.1 kcal mol^−1^. More details on the speciation of these intermediates can be found in Appendix A. For this set of dimers (**8** to **9^MeO(t),n = 0^**), the closed-shell singlet state corresponding to the mixed-valence dimer Cu(III)-Cu(I) is always computed higher in energy.

The triplet Cu(II)-Cu(II) complex **9^MeO(t),n = 0^** can then undergo Cu-Cu transmetalation to yield the mixed-valence (Cu(III)-Cu(I)) singlet complex **10^MeO(s),n = 0^**. This latter is not a true minimum on the potential energy surface and spontaneously undergoes reductive elimination to yield Cu(I)-Cu(I) complex **11^MeO(s),n = 0^** and biphenyl (Appendix A). Since this transformation requires a spin transition, the associated singlet and triplet energy surface were scanned to estimate the activation energy of this process. The electronic energy of activation for the transmetalation and coupled electron transfer (TET) was estimated to 5 and 7 kcal mol^−1^. In the case of the monosolvated **10^MeO,n = 1^**, the barrier for the TET process is nearly doubled and the product of transmetalation complex **11^MeO(s),n = 1^** is an energy minimum (Appendix A). Note that direct reductive elimination at monomeric species Cu(II) is kinetically prohibited (Appendix A). This mechanistic picture for reductive elimination is in agreement with results recently reported by Casares on the role of electron transfer between organocopper complexes in transmetalation reactions [49].

## 3. Conclusions

The presence of a coordinating base (hydroxide and methoxide) enables the first B-to-Cu(II) transmetalation from aryl boronic acid to Cu^II^Cl_2_ in methanol, while an excess of base is detrimental (Figure 3). This suggests the formation of heterobimetallic Cu-(μ-OH)-B intermediates as previously suggested [45,50] and as confirmed by our DFT calculations. In contrast with previous suggestions from the literature, a second B-to-Cu transmetalation to form bis-aryl Cu(II) complexes is kinetically prevented [6,7,9]. Alternatively, the formation of organocopper(II) dimers is favored and the latter can undergo a coupled transmetalation-electron transfer (TET) allowing the formation of a mixed-valence organocopper(III)/copper(I) complex able to perform reductive elimination.

The possibility to catalyze homocoupling reactions using Cu(I) in the absence of a base suggests that transmetalation proceeds either via Cu(I) oxidation by O_2_ and the associated generation of OH^−^ [51,52]:2 Cu^+^ + 1/2 O_2_ + H_2_O = 2 Cu^2+^ + 2 OH^−^
or through a mechanism involving a Cu(I) oxo complex as proposed in the case of Pd(0) or via another oxidation intermediate [27,28]. This point is currently under study within our group.

Based on mechanistic insights herein reported, we can tentatively propose some guidelines to control the undesired formation of homocoupling product in the context of Cu-catalyzed cross-coupling reactions. The pH should be controlled and optimized for each catalytic system to favor transmetalation and prevent the side reactions. The use of bidentate ligands is expected to disfavor the formation of dimeric Cu(II) species responsible for homocoupling. Alternatively, using a supported Cu catalyst or working at low catalyst loadings may also be helpful. Finally, the control of the oxygen concentration in the media is crucial to limit the formation of Cu(II) species in Cu-catalyzed non-oxidative cross-coupling reactions.

## 4. Materials and Methods

Structure optimizations, energy estimation, and NBO 6 analyses [53] were performed with the gaussian 09 software [54]. Molecules have been optimized using an ultrafine grid as implemented in gaussian. The Becke hybrid functional, B3LYP functional was used for all calculations [35,36,37,38]. The 6–31+G augmented double-ζ Pople’s basis sets were used for B, C, and H atoms, and complemented with an extra polarization function for O atom. Quasi relativistic Stuttgart–Dresden 10MWB electron core pseudopotentials and their associated basis sets were used for Cu and Cl atoms [55]. Since the reaction occurs in a highly polar solvent, particular attention was paid to model the solvation. The most favorable solvation states of key intermediates have been verified (see the Appendix A), and the bulk solvation effect has been represented by a continuum model for all calculations [56]. Unless specified, optimized transition states were checked by analytical frequency calculations and their connectivity were verified by IRC following.

## Data Availability

The data that support the findings of this study are available in the Figures or Appendix A of this article.

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
