# Peer review of "Copper-Catalyzed Homocoupling of Boronic Acids: A Focus on B-to-Cu and Cu-to-Cu Transmetalations"

_molecules, 2022, doi:10.3390/molecules27217517_

Round 1
Reviewer 1 Report
The paper is written very well, it is clear and concise. The authors explain very well about copper catalyzed homocoupling of boronic acids to form the C-C bond formation. I recommended to accept without any modification.
Author Response
We thank this reviewer for his / her warm consideration for our work.
Reviewer 2 Report
The authors presnet their understanding of the copper-catalyzed homocoupling of boronic acids which involves the transmetallization of Cu-to-Cu frong Cu-to-B intermediates. This is a very interesting and significant for the development of effecitive Cu-catalyzed cross-coupling reactions. Both the experimental results and the theoretical caculations support the conclusion very well. The manuscript is publishable. The only comment is as followed:
In Scheme 3, the side product other than the coupling product is C6H5F, which is actually the radical FC6H4 gaining a H from the solvent CH3OH. Why isn't the coulping product produced by the direct coupling between the two FC6H4 radicals, but by such a non-stable tetra-atomic ring containing two Cu atoms, which undergoes a intermolecular rearrangement and electron transfer between the two Cu atoms?
Author Response
We thank this reviewer for his / her warm consideration for our work.
Regarding the formation of C6H5F, we propose that it takes place via the protonation of the intermediate [Cu(II)]-C6C4F by MeOH (to regenerate [Cu(II)(OMe)] and C6H5F) rather than through a radical mechanism (given that the formation of the MeO° radical would be highly unfavorable).
The direct coupling between two °C6H5F radical moieties could be an alternative path but given the high energy cost of forming an aryl radical the probably that two of them would encounter to couple is very low.
Reviewer 3 Report
In this work, detailed experiments and DFT calculations on copper-Catalyzed homocoupling of boronic acids were displayed by authors both on B-to-Cu and Cu-to-Cu transmetalations that the presence of a coordinating base (hydroxide, methoxide) enables the B-to-Cu(II) transmetalation from aryl boronic acid to CuIICl2 in methanol, through the formation of a mixed Cu-(μ-OH)-B intermediates; the formation of organocopper(II) dimers that later can undergo a coupled transmetalation-electron-transfer (TET) to form the mixed-valence organocopper(III)/copper(I) complex which promotes reductive elimination. Finally, some guidelines to control the undesired formation of homocoupling products in the context of Cu-catalyzed cross-coupling reactions were suggested. Therefore, this reviewer would like to recommend this manuscript be published on molecules.
Author Response

(The authors gave the same response as above.)
